# Indoxyl Sulphate Retention Is Associated with Microvascular Endothelial Dysfunction after Kidney Transplantation

**DOI:** 10.3390/ijms24043640

**Published:** 2023-02-11

**Authors:** Sam Hobson, Samsul Arefin, Awahan Rahman, Leah Hernandez, Thomas Ebert, Henriette de Loor, Pieter Evenepoel, Peter Stenvinkel, Karolina Kublickiene

**Affiliations:** 1Division of Renal Medicine, Department of Clinical Science, Intervention and Technology, Karolinska, Institutet, 141 52 Stockholm, Sweden; 2Medical Department III—Endocrinology, Nephrology, Rheumatology, University of Leipzig Medical Center, D-04103 Leipzig, Germany; 3Nephrology and Renal Transplantation Research Group, Department of Microbiology, Immunology and Transplantation, Katholieke Universiteit Leuven, BE-3000 Leuven, Belgium; 4Department of Nephrology and Renal Transplantation, University Hospitals Leuven, BE-3000 Leuven, Belgium

**Keywords:** chronic kidney disease, EndoPAT, endothelial dysfunction, indoxyl sulphate, kidney transplantation, nitric oxide, p-cresyl sulphate, vessel stiffness

## Abstract

Kidney transplantation (KTx) is the preferred form of renal replacement therapy in chronic kidney disease (CKD) patients, owing to increased quality of life and reduced mortality when compared to chronic dialysis. Risk of cardiovascular disease is reduced after KTx; however, it is still a leading cause of death in this patient population. Thus, we aimed to investigate whether functional properties of the vasculature differed two years post-KTx (postKTx) compared to baseline (time of KTx). Using the EndoPAT device in 27 CKD patients undergoing living-donor KTx, we found that vessel stiffness significantly improved while endothelial function worsened postKTx vs. baseline. Furthermore, baseline serum indoxyl sulphate (IS), but not p-cresyl sulphate, was independently negatively associated with reactive hyperemia index, a marker of endothelial function, and independently positively associated with P-selectin postKTx. Finally, to better understand the functional effects of IS in vessels, we incubated human resistance arteries with IS overnight and performed wire myography experiments ex vivo. IS-incubated arteries showed reduced bradykinin-mediated endothelium-dependent relaxation compared to controls via reduced nitric oxide (NO) contribution. Endothelium-independent relaxation in response to NO donor sodium nitroprusside was similar between IS and control groups. Together, our data suggest that IS promotes worsened endothelial dysfunction postKTx, which may contribute to the sustained CVD risk.

## 1. Introduction

Over 800 million people currently live with chronic kidney disease (CKD), a leading cause of mortality worldwide [1]. This figure is only expected to rise in the future and is predicted to become the fifth highest cause of mortality by 2040 [2]. Of note, 40–50% of deaths in advanced CKD patients are cardiovascular disease (CVD)-related [3]. Traditional risk factors are not sufficient to solely account for this increased risk; thus, additional non-traditional risk factors such as uremic toxin retention [4], hyperphosphatemia and inflammation have been attributed [5]. Early vascular ageing (EVA) ensues, defined as a discrepancy between chronological and biological age in the vasculature [6], and is characterized by increased vessel stiffness, vascular calcification and endothelial dysfunction [7]—all of which are independent risk factors for CVD in CKD [8,9,10]. Indeed, kidney transplantation (KTx) is the preferred renal replacement therapy for end-stage kidney disease patients both in the terms of survival and quality of life [11]. Nonetheless, kidney transplant recipients still experience an elevated risk of cardiovascular-related death postKTx [12], which cannot be explained by the presence of traditional risk factors [13].

Endothelial dysfunction describes the change from physiological to pathological activity in endothelial cells, characterized by prothrombotic activity, reduced vasodilation and a proinflammatory state [14]. It is observed in small and large conduit arteries in CKD patients [15,16] and is associated with increased risk of cardiovascular-related mortality in CKD [17]. Non-traditional risk factors such as uremic toxin accumulation and oxidative stress have also been linked with endothelial dysfunction. Moreover, vascular smooth muscle cell (VSMC) structure and function, together with extracellular matrix composition, play an important role in vessel stiffening (arteriosclerosis)—a process related to changes in composition of elastin, collagen and vascular calcification, primarily in the tunica media layer [18].

The effects of KTx on vascular function have been previously studied. While several studies suggest that KTx ameliorates vessel stiffness [19,20], the jury is still out regarding whether KTx improves or worsens endothelial function [21,22]. Many of these investigations have assessed vascular function in larger arteries; hence, less is known about the microcirculation, shown to be severely impaired in patients with CKD [23]. Gut-derived uremic toxins, p-cresyl sulphate (pCS) and indoxyl sulphate (IS), are protein-bound uremic toxins that promote glomerular sclerosis and interstitial fibrosis in the kidney through oxidative stress and inflammatory reactions [24]. IS is associated with vascular disease and mortality in CKD patients [25], while both solutes have been associated with vascular malfunction [26,27,28]. Impaired endothelium-dependent dilatation in uremic conditions has previously been demonstrated [29], but further studies are needed to investigate the potential detrimental effects of these toxins in the microcirculation and determine which pathways are impaired.

In the present study, we hypothesized that living donor-KTx (LD-KTx) has beneficial effects on endothelial function and vessel stiffness in the microcirculation, assessed using EndoPAT—a device previously reported to identify patients with early coronary atherosclerosis and CVD [29,30]. Therefore, we compared functional and biochemical markers of endothelial function and vascular stiffness in patients at KTx (baseline) vs. two years post living-donor KTx (postKTx). We then investigated their associations with pCS and IS at baseline, studied concentrations of markers of vascular remodeling matrix metalloproteinase 9 (MMP-9) and P-selectin at baseline vs. postKTx [31,32] and finally tested ex vivo effects of IS on resistance arteries to gain mechanistic insights of IS-induced vascular dysfunction.

## 2. Results

### 2.1. Characteristics of Study Population

Characteristics of patients at baseline and postKTx are presented in Table 1. All participants studied at baseline were present at follow-up. Of these, 78% of patients were male, while the median age at baseline was 45 years. Body mass index (BMI), statin usage, calcium, high-density lipoprotein (HDL) cholesterol and coronary artery calcium (CAC) score all significantly increased postKTx compared to baseline, while systolic/diastolic blood pressure, albumin, phosphate and lipoprotein(a) significantly decreased postKTx. The study design is depicted in Figure 1.

### 2.2. In Vivo Functional Study

Firstly, we investigated the long-term effects of LD-KTx on components of EVA using EndoPAT. Reactive hyperemia index (RHI) was decreased postKTx compared to baseline in 20/27 (74%) patients (median (IQR): 2.18 (1.63–2.50) vs. 2.51 (2.17–2.91); *p* < 0.01; Table 1, Figure 2A), reflecting worsened endothelial function after KTx. RHI < 1.67 and < 2.08 have previously been shown to discriminate between patients at increased risk of coronary artery disease and chest pain, respectively [29,33]. In our study, two patients had a RHI < 1.67 at baseline, while seven were below the threshold postKTx. Similarly, 6 CKD patients had a RHI < 2.08 at baseline and 13 postKTx.

On the contrary, augmentation index (AIx) was attenuated postKTx compared to baseline in 21/27 (78%) patients (median (IQR): −6% (−17–2) vs. 4% (−7–10); *p* < 0.01; Table 1, Figure 2B), reflecting amelioration of vessel stiffness after KTx.

### 2.3. Biochemical Parameters before and after KTx

Next, we investigated the effects of renal function restoration on two novel biomarkers of vascular function: P-selectin and matrix metalloproteinase 9 (MMP-9). No difference was observed for plasma P-selectin at baseline *vs.* postKTx (median: 43.0 (37.3–56.5) ng/mL *vs.* 44.55 (37.3–52.3) ng/mL; *p* > 0.05; Table 1). Similarly, no difference was observed for serum MMP-9 at baseline *vs.* postKTx (median: 515.1 (368.3–756.0) ng/mL *vs.* 479.0 (327.9–784.3) ng/mL; *p* > 0.05; Table 1).

### 2.4. Univariate Correlations and Multivariate Regression Analyses

IS and pCS are protein-bound uremic toxins that have cytotoxic effects in multiple tissues leading to detrimental outcomes, including the endothelium. Therefore, we quantified serum concentrations at baseline and looked at their relationship with other markers of EVA reported above. We found IS inversely correlated with RHI (ρ: −0.461; *p* < 0.05; Table 2) and positively correlated with *P*-selectin (ρ: 0.510; *p* < 0.01; Table 2), both at postKTx only. Correlations between pCS and all other biomarkers of vessel remodeling were not significant (all *p*-values > 0.05; Table 2).

To identify independent associations between uremic toxins and markers of endothelial function or vessel stiffness, we performed multivariate linear regression analyses with an adjustment for age, sex, presence of diabetes, dialysis treatment and statin usage, only for those parameters that were significantly correlated to IS or pCS in univariate analyses. Our analyses revealed that IS was independently negatively associated with RHI postKTx (standardized beta = −0.506; *p* < 0.05; Table 3). IS was also independently positively associated with P-selectin postKTx (standardized beta = 0.518; *p* < 0.05; Table 3).

A multivariate model was calculated only for correlations for which a significant univariate correlation was found (Table 2). Non-normally distributed variables were log10 transformed prior to analysis. Blood pressure-lowering medication was not included since all patients were on this treatment at baseline. Standardized ꞵ coefficients and *p*-value are given for each model. Significant associations (*p* < 0.05) after adjustment for covariates are depicted in **bold**.

### 2.5. Ex Vivo Functional Study

Finally, considering the above findings between IS and RHI/P-selectin postKTx, we investigated the ex vivo effects of IS in an isolated organ culture bioassay. Briefly, isolated resistance arteries from non-CKD individuals were cultured with 100 µM IS for 24 h before vascular function and structure were assessed using wire myography. The median (interquartile range) age and BMI of the participants were 44 (35–47) years and 25.9 (22.9–34–7) kg/m^2^, respectively; 38% of subjects were male.

Resistance arteries from the IS-incubated group (n = 6) showed reduced bradykinin (BK)-mediated endothelium-dependent relaxation compared to the control group (*p* < 0.001; Figure 3A). When we investigated the contribution of different mediators of endothelium-dependent dilation, i.e., nitric oxide (NO) and endothelium-derived hyperpolarizing factor (EDHF), we observed that NO contribution was significantly reduced in the IS group compared to the control group (*p* < 0.05; Figure 3B); however, there was no difference in EDHF contribution between control and IS groups (Figure 3B). Endothelium-independent relaxation stimulated from NO donor SNP was unchanged (*p* > 0.05; Figure 3C). Lastly, endothelium-derived vasodilator prostacyclin (PGI2) did not contribute to the vasodilatory response in small resistance vessels, as shown by similar responses to BK with and without indomethacin (Figure 3D).

## 3. Discussion

Supportive evidence suggests that the earliest manifestations of EVA occur at the level of the microcirculation [34], while endothelial dysfunction has been observed in patients of all age groups, from the earliest stages of CKD [35] to advanced CKD [23]. However, knowledge surrounding changes to vascular structure and function after KTx is lacking. Further research is necessary to determine vascular changes and the mechanisms behind them, which may tailor screening and treatment to reduce CVD risk in CKD patients [36]. In the present study, we report that, in contrast to our expectations, endothelial function worsens while AIx improves postKTx compared to baseline in the defined CKD patient group undergoing LD-KTx. These changes are related to the peripheral microcirculation assessed using EndoPAT. Differences in biochemical markers related to increased CVD risk were not observed after KTx, but IS is independently negatively associated with RHI and independently positively associated with P-selectin, both at postKTx. To support this finding, using isolated small artery bioassays, we showed that IS has a detrimental effect on resistance artery endothelial function, confirmed by impaired dilatation and a preserved dilatory response to NO donor SNP. This strengthens the notion that certain uremic toxins detrimentally affect vascular maintenance and function after KTx.

KTx has superior long-term survival compared to chronic dialysis [37]. The CVD burden in CKD patients is extremely high compared to the healthy age-matched population; however, CVD still accounts for 30% of hospitalizations in postKTx patients [38]. Traditional and non-traditional risk factors contribute to the increased risk, which may be present before KTx, such as obesity [39], or commence after KTx, such as post-transplant diabetes or dyslipidemia [12,40]. Other important factors include the presence of hypertension [41] which, together with the uremic environment, may stimulate multiple pathogenic mechanisms such as activation of the renin-angiotensin-aldosterone system, sympathetic hyperactivity and sodium retention [42]. In addition, endothelial cell senescence, macro-/microvascular ageing and increased angiogenic factors may amplify each other’s effects towards aggravation of microvascular disease [34]. Taken together, it was unclear whether endothelial function of the microcirculation would improve postKTx.

Studies investigating the effects of KTx on endothelial function have been conducted before, but the data are inconsistent regarding whether function improves [21,43,44], worsens [22] or is preserved [45] over time. Discrepancies in the vascular bed investigated, patient population with pre-existing comorbidities and/or follow up period after KTx may explain these differences. Indeed, several studies failed to monitor vascular function in suitably matched subjects over one year postKTx [46,47] or used a methodology inferior to EndoPAT to test endothelial function, such as flow-mediated vasodilation (FMD) [43,48] or biomarkers reflecting endothelial damage, such as intercellular adhesion molecule 1 (iCAM-1) and vascular adhesion molecule 1 (vCAM-1) [45]. The EndoPAT device is a suitable method to comprehensively assess functional aspects of EVA and has many advantages over older FMD methods; EndoPAT is both operator-independent and able to control for autonomous nervous system tone [49]. Moreover, despite using a similar reactive hyperemia technology to assess endothelium function, FMD assesses conduit artery function, while EndoPAT evaluates microvascular beds—an important consideration taking into account their differences in structure and function [50]—and the importance of the microcirculation as a treatment option in CKD patients [34].

The detrimental effect of IS on endothelial function ex vivo was in line with our in vivo findings and was supported by an independent association between IS concentrations at baseline and RHI postKTx after adjustment for clinically relevant covariates, i.e., higher IS levels at baseline were predictive of worsened endothelial function postKTx (findings illustrated in Figure 4). A comparable trend was observed between IS and RHI at baseline in univariate correlation analysis, but this did not reach statistical significance. Although we acknowledge that serum concentrations of uremic toxins decrease following the restoration of renal function [51], we did not have the possibility to measure IS/pCS postKTx. Using pharmacological tools to inhibit NOS/COX pathways, we showed that the NO pathway is impaired in arteries incubated with IS. The vasodilatory response was preserved in IS-incubated and control arteries stimulated with SNP (endothelium-independent relaxation), suggesting impairment is upstream of the NO machinery, and the pathway downstream of NO is functioning, i.e., VSMCs are tolerant to IS. IS-induced oxidative stress may explain reduced NO contribution, since previous experimental data showed that IS inhibits NO production by inducing oxidative stress in vascular endothelial cells [52,53]. Dou et al. demonstrated that IS not only induces oxidative stress through increased NADPH oxidase activity from endothelial cells but also through decreased glutathione, a scavenger of hydrogen peroxide and hydroxyl radical [54]. Another endothelium-derived mediator, EDHF, was not affected after IS incubation and did not show a compensatory response, further explaining overall reduced endothelial function after IS incubation. Our findings are also supported by Takayuki et al. who showed that IS caused direct impairment of endothelial function through reduced NO, and not EDHF or PGI2, in rat superior mesenteric arteries [55].

In contrast to RHI, AIx significantly increased postKTx vs. baseline, suggesting improved structural improvement after KTx, and may partially explain reduced CVD risk in comparison to CKD patients. Our data are in line with several other studies that used either AIx or pulse wave velocity to determine vessel stiffness before and after KTx at different timepoints, ranging from two weeks to one year postKTx [48,56,57,58,59], in comparison to our two-year study. Thus, our data not only confirm previous findings but also add new insight into improvement of vessel stiffness over a longer period postKTx. AIx is elevated in patients with increased peripheral resistance, i.e., the reflected pulse wave is returned at a quicker speed [60]. It is increased in hypertension and influenced by age, height, heart rate and biological sex [48]. In addition, change in mean arterial pressure after KTx is an independent risk factor for change in AIx [61] and graft quality may influence post-transplant vascular outcomes [62]. We observed significantly reduced systolic/diastolic blood pressure, albumin, phosphate and lipoprotein(a), as well as increased body mass index (BMI), statin usage, calcium, high-density lipoprotein (HDL) cholesterol and coronary artery calcium (CAC) after postKTx—all of which may partially explain, or explain in composite, observed changes to endothelial function and/or vessel stiffness; further and larger studies are required for clarification. To our surprise, CAC score increased after KTx; however, increased CAC and aortic calcification have been previously observed 3.5–4.4 years postKTx [63,64].

Inclusion of a homogenous CKD patient population undergoing LD-KTx is a major strength of our study. We also used a range of techniques to better understand and validate mechanisms by which the endothelium responds to KTx. We also acknowledge limitations in our study, such as only implementing one follow-up visit into the study design, which precludes studying the initial short-term response of KTx. In addition, it would be of great interest to assess IS concentrations after KTx to correlate with markers of vascular remodeling after two years, but it was not possible in the current study. Lastly, we were unable to stratify patients by use of medication due to the small sample size and only collecting data at two time points. We were, however, able to partly resolve this issue by adjusting for statin use at baseline in our multivariate linear regression models.

In summary, we report that endothelial function worsened while vessel stiffness improved postKTx. IS was independently associated with markers of endothelial dysfunction postKTx and IS-induced endothelial damage was associated with reduced NO contribution ex vivo, while EDHF contribution was preserved. By strengthening evidence of IS as an “endothelial toxin”, novel therapeutics that target circulating IS may help reduce CVD burden in postKTx patients. Further studies are needed to determine relationships between endothelial function and vessel stiffness in this patient population.

## 4. Materials and Methods

### 4.1. Patients and Study Design

The present study focuses on CKD5 patients (n = 27) undergoing living-donor KTx at the Department of Transplantation Surgery at the Karolinska University Hospital. In this longitudinal study, all study participants were sampled immediately before KTx and two years postKTx before comparisons were made, including EndoPAT measurements, blood samples for future analysis of pre-specified markers and blood pressure readings. Basic patient characteristics are outlined in Table 1. All patients were invited to participate in the study and provided written informed consent. To investigate the effects of soluble IS on vascularity reactivity, isolated arteries from non-CKD participants undergoing planned inguinal hernia repair, cholecystectomy, bariatric surgery or kidney transplantation (donors) were used. The Swedish Ethical Review Authority in Stockholm approved the study, and all subsequent experiments were performed in accordance with the Declaration of Helsinki.

### 4.2. EndoPAT

The Endo-PAT 2000 device (Itamar Medical, Caesarea, Israel) assessed RHI and AIx representing endothelial function and arterial stiffness, respectively. The test was performed by a trained investigator following the manufacturer’s instructions. In brief, participants were asked to refrain from talking and remain in a relaxed state for the duration of the study. While lying in a supine position, probes were inflated around the participants’ fingers and a blood pressure cuff was placed around the bicep of the test arm before both forearms were placed on arm supports. Additional foam anchor rings and tape were used to ensure the probes did not touch stray fingers or the arm supports. The examination consisted of three 5 min recordings. First, a baseline period of 5 min was recorded before blood flow in the brachial artery of the test arm was occluded for 5 min by rapidly inflating the blood pressure cuff to at least 60 mmHg above systolic blood pressure and no less than 200 mmHg. Finally, the occlusion was released by deflating the pressure cuff for a 5 min post-occlusion recording period. The EndoPAT software automatically calculated RHI and AI. RHI is quantified as a ratio of post-occlusion to pre-occlusion pressure in the test arm, divided by the ratio of post-occlusion to pre-occlusion pressure in the control arm. The AIx was calculated from the difference between the systolic peaks of the PAT waveform. The heart rate-corrected AIx (AIx@75) was used for further analysis [65].

### 4.3. Biochemical Analysis/Clinical Parameters

Blood specimens were routinely drawn at the time of KTx and postKTx (two years after). Subjects were fasted at the time of blood collection. Samples were stored for <2 h at 4 °C until centrifugation. Upon arrival at the laboratory, the blood samples were centrifuged at 2650G for 20 min, aliquoted and either processed immediately (standard techniques) or stored at −80 °C until analysis. Creatinine, albumin, calcium, phosphate, total cholesterol, triglycerides, HDL cholesterol and high-sensitivity C-reactive protein (hsCRP, high sensitivity nephelometric assay) were measured using standard methods at the Department of Laboratory Medicine, Karolinska University Hospital at Huddinge [66]. Concentrations of total IS and pCS were centrally quantified from patient serum with a previously described method [67]. In brief, serum samples were deproteinized with acetonitrile after addition of an internal standard (stable isotope labeled analogues) and then filtered over a 96-well Ostro plate (Waters, Zellik, Belgium). After drying with nitrogen and redissolving in MilliQ water, the samples were analyzed using ultra-performance liquid chromatography—tandem mass spectrometry with negative electrospray ionization. Relevant demographics, comorbidities, medications and routine biochemistry were extracted from electronic files. Blood pressure-lowering medication includes use of beta-blockers, diuretics, angiotensin-converting enzyme inhibitors, angiotensin II receptor blockers and/or calcium channel blockers.

Two biomarkers of vascular remodeling, P-selectin and MMP-9, were analyzed in plasma or serum using enzyme-linked immunosorbent assay (ELISA) kits. Human P-selectin Quantikine ELISA kit (RAB0426-1KT; Sigma-Aldrich, St. Louis, MO, USA) and Human MMP-9 Quantikine ELISA kit (DMP900; R&D Systems, MA, USA) were performed according to the manufacturers’ instructions.

### 4.4. Vascular Reactivity Studies in Resistance Arteries

The median (interquartile range) age and BMI of the participants were 44 (35–47) and 25.9 (22.9–34–7), respectively, while 38% of participants were male. During planned operations, a piece of subcutaneous fat was removed at the incision site. Samples were immediately placed in iced physiological salt solution (PSS) and kept at 4 °C until dissection. Resistance arteries (diameter ≈ 150–400 µm) were dissected from the sample using stereomicroscopy. One or two artery segments (segment length ≈ 1.6–2.0 mm) were isolated from each participant depending on the number of arteries available in the fat biopsy. Isolated arteries were cultured in wells containing DMEM medium supplemented with 5% FBS, 1% Na Pyruvate, 2.5% penicillin/streptomycin, 1% FungiZone, 1% L-glutamine and 1% HEPES at 37 °C in a humidified atmosphere with 5% CO_2_ for 24 h [68], in the presence or absence of 100 µM IS (concentration used based on the median total IS concentration observed in our cohort (100.5 µM, Table 1)). Vessels were mounted into organ baths of a Multi-Myograph System (model 610M, Danish Myo Technology A/S, Denmark). The organ baths were perfused with PSS, heated to 37 °C and aerated constantly with 95% O₂ and 5% CO₂. After an equilibration phase of 60 mins, a standardized normalization protocol was performed to determine the optimal circumference. Inclusion criteria for vessel segments was >50% endothelium-dependent relaxation with ACh (10 µM) or BK (1 µM) after pre-contraction with phenylephrine (10 µM) and force development of at least 1 mN/mm for each vessel. Firstly, a concentration–response curve was established for norepinephrine (NE) (1 nM–3 µM). Once a sustained, steady contraction to NE (3 µM) was attained, concentration–response curves for endothelium-dependent vasodilator BK (10 pM to 1 µM) were obtained. To assess the contribution of endothelium derived factors, arteries were then incubated with nitric oxide synthases (NOS) inhibitor L-NG-Nitro arginine methyl ester (L-NAME, 100 µM) and cyclooxygenase (COX) inhibitor indomethacin (100 µM) for 20 mins and pre-constricted again to establish a concentration–response curve for BK (10 pM to 1 µM). For endothelium-independent dilation, a concentration–response curve for endothelium-independent vasodilator SNP (NO-donor) (1 nM to 100 µM) was obtained. At the end of each experiment, a stretching procedure in the presence of 1 mM SNP, 0.2 mM papaverine and 1 mM EGTA in Ca2+ free PSS was completed to obtain the passive-length relationship.

A total of 6 participants were included in the IS incubation group, and 8 participants were included in the control group. For each experimental condition, i.e., control and IS, only one vessel segment was used; no repetition or duplicates were performed. BK and SNP relaxation were calculated as % relaxation achieved, whereby the stable level of contraction prior to the addition of vasodilators was set at 0% and the baseline obtained before the addition of contractile stimuli as 100%. The contribution of NO and EDHF was then calculated. Since NOS and COX inhibition blocks NO and PGI2, the remaining response was considered to be EDHF-mediated dilatation. Of note, differences in dilation after COX inhibition were not observed (Figure 3D) suggesting a PGI2 response is absent in resistance arteries; thus, we only considered NO and EDHF to be present. By subtracting the EDHF response from the overall vasodilatory response, we calculated the NO contribution.

### 4.5. Statistical Analysis

Statistical analysis was performed using GraphPad Prism v8.0.2 (San Diego, CA, USA) SPSS Statistics 29.0 (Armonk, NY, USA) and STATISTICA 7.0 (StatSoft, Uppsala, Sweden). For all continuous variables, Kolmogorov–Smirnov test was used to test for normality. The paired-sample *t*-test or Wilcoxon matched-pairs signed rank test was used to compare differences between continuous variables with repeated measures, while McNemar’s test was used for categorical variables. Univariate correlations were assessed using Spearman’s rank correlation. To identify independent associations between IS and markers of endothelial dysfunction or vessel stiffness, multivariate linear regression models for significant correlations in univariate analyses were used. Covariates in the multivariate models included age at study entry, sex, presence of diabetes, dialysis treatment and statin usage at baseline. In all multivariate models, non-normally distributed variables were log10 transformed. Missing values: Lipoprotein(a) = 3, AGE score = 1, HDL cholesterol = 1, IS = 1, pCS = 1, P-selectin = 1. For concentration-dependent ex vivo experiments, groups were compared using two-way ANOVA. Differences between continuous variables were assessed using Mann–Whitney U test. *p*-values < 0.05 were considered statistically significant.

## Figures and Tables

**Figure 1 ijms-24-03640-f001:**
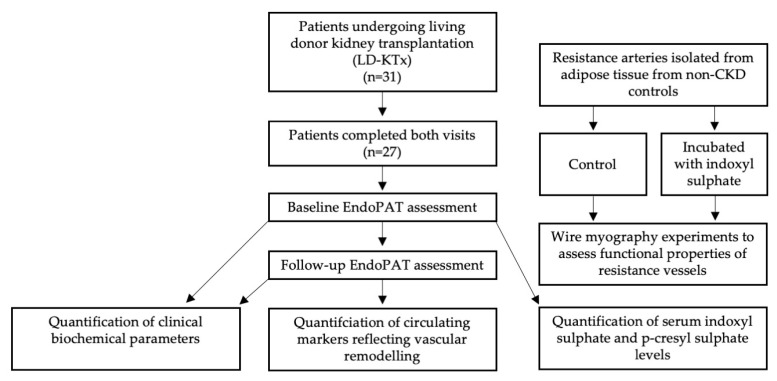
Flowchart depicting study design.

**Figure 2 ijms-24-03640-f002:**
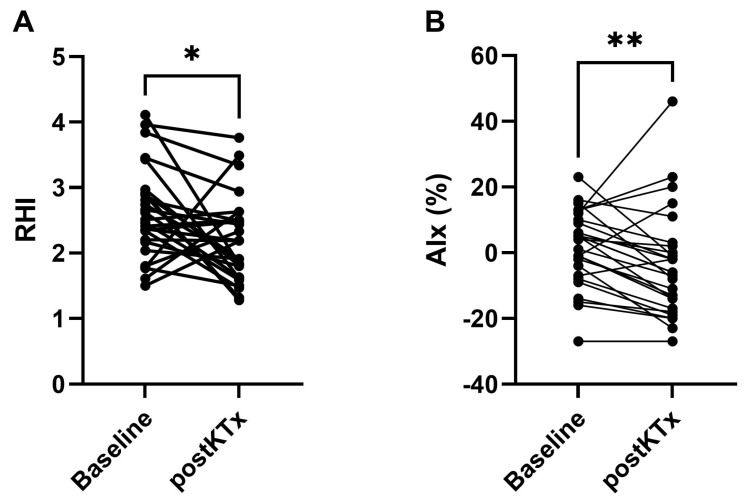
Reactive hyperemia index (**A**) and augmentation index (**B**) at baseline and two years post-kidney transplantation (postKTx) (n = 27) assessed using the EndoPAT device. Differences between baseline and postKTx were assessed using paired *t*-test or Wilcoxon matched-pairs signed rank test. Significant *p*-values are shown above graphs; * < 0.05, ** < 0.01. Abbreviations: postKTx, two years post-transplantation; AIx, augmentation index; RHI, reactive hyperemia index.

**Figure 3 ijms-24-03640-f003:**
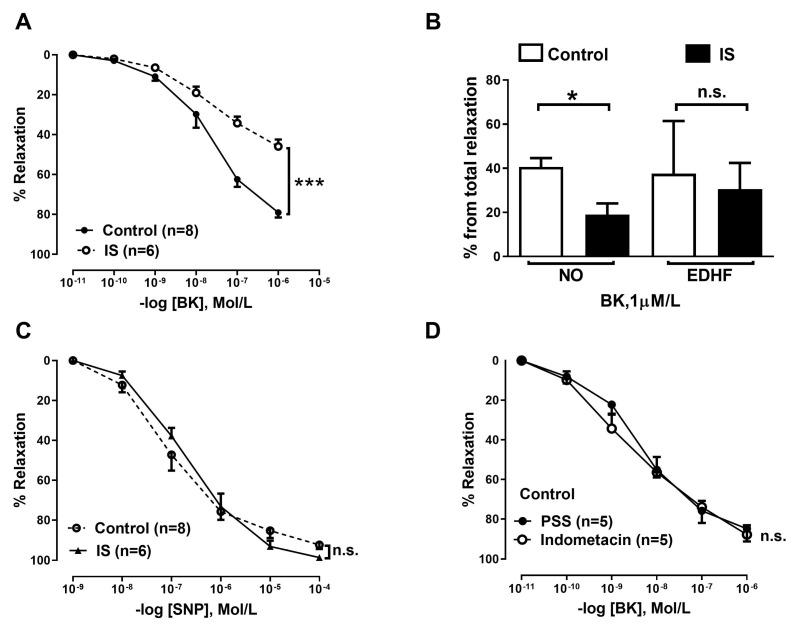
Indoxyl sulphate-induced endothelial dysfunction in resistance arteries is associated with reduced NO contribution. (**A**) Concentration-response curves for BK-induced relaxation in vessels preincubated with IS (n = 6) for 24 h compared to control arteries (n = 8); (**B**) contribution of NO and EDHF in IS and control groups; (**C**) concentration-response curves for SNP-induced vasodilation for control and IS groups; (**D**) concentration-response curve for BK-induced relaxation in resistance arteries before and after indomethacin incubation. Vessel relaxation from BK and SNP were calculated as % relaxation whereby the stable level of contraction prior to the addition of vasodilators was set at 0% and the baseline obtained before the addition of contractile stimuli as 100%. Data are expressed as mean ± SEM (**A**,**C**,**D**) or median (interquartile range) (**B**). Differences were assessed by two-way ANOVA (**A**,**C**,**D**) or Mann–Whitney U test (**B**). * *p* < 0.05, *** *p* < 0.001. Abbreviations: BK, bradykinin; endothelial-derived hyperpolarizing factor, EDHF; IS, indoxyl sulphate; SNP, sodium nitroprusside.

**Figure 4 ijms-24-03640-f004:**
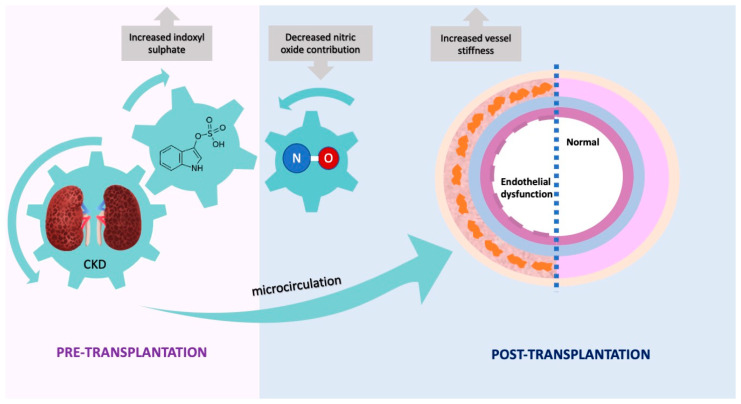
Proposed model of indoxyl sulphate-induced endothelial dysfunction in the microcirculation after kidney transplantation.

**Table 1 ijms-24-03640-t001:** Patient characteristics of the study cohort at baseline and two years post-kidney transplantation.

	Baseline (N = 27)	postKTx (N = 27)	*p*-Value
Age (years)	45 (35–54)	48 (31–55)	-
Men	21 (78%)	21 (78%)	-
BMI (kg/m^2^)	24.5 (23.0–28.7)	25.4 (24.6–29.4)	**0.019**
SBP (mmHg)	138 (128–158)	130 (119–137)	**0.005**
DBP (mmHg)	90 (81–98)	82 (78–86)	**0.032**
Diabetes mellitus	1 (4%)	2 (7%)	1.000
Lipid-lowering medication	10 (37%)	23 (85%)	**<0.001**
Blood pressure-lowering medication	27 (100%)	25 (93%)	0.500
Dialysis treatment	7 (26%)	-	-
eGFR (mL/min/1.73 m^2^)	-	57 (41–69)	-
Albumin (g/L)	38 (36–39)	36 (32–38)	**0.023**
hsCRP (mg/L)	1.4 (0.4–2.6)	0.9 (0.5–3.0)	0.976
Calcium (mmol/L)	2.3 (2.1–2.4)	2.4 (2.3–2.5)	**0.018**
Phosphate (mmol/L)	1.7 (1.5–2.0)	0.9 (0.8–1.0)	**<0.001**
Total chol (mmol/L)	4.4 (3.5–5.5)	4.1 (3.9–4.4)	0.727
HDL cholesterol (mmol/L)	1.3 (1.0–1.5)	1.6 (1.3–1.8)	**<0.001**
Triglycerides (mmol/L)	1.8 (1.0–2.0)	1.4 (0.9–1.7)	0.078
Lipoprotein(a) (mg/L)	18 (10–76)	10 (10–37)	**0.019**
Indoxyl Sulphate (µmol/L)	100.8 (82.8–153.5)	-	-
p-Cresyl Sulphate (µmol/L)	191.6 (101.5–281.5)	-	-
CAC score (AU)	3 (0–61)	27 (0–125)	**0.001**
AGE score	3.0 (2.4–3.4)	-	-
RHI	2.51 (2.17–2.91)	2.18 (1.63–2.50)	**0.019**
AIx (%)	4 (−7–10)	−6 (−17–2)	**0.009**
P-selectin (ng/mL)	43.0 (37.3–56.5)	44.55 (37.3–52.3)	0.582
MMP-9 (ng/mL)	515.1 (368.3–756.0)	479.0 (327.9–784.3)	0.604

postKTx, post-transplantation; AGE, advanced glycated end-product; AIx, augmentation index; BMI, body mass index; CAC, coronary artery calcification; chol., cholesterol; DBP, diastolic blood pressure; eGFR, estimated glomerular filtration rate; hsCRP, high sensitivity C-reactive protein; MMP-9, matrix metalloproteinase 9; RHI, reactive hyperemia index; SBP, systolic blood pressure. Blood-pressure-lowering medication includes use of beta-blockers, diuretics, angiotensin-converting enzyme inhibitors, angiotensin II receptor blockers and/or calcium channel blockers. Data are presented as median (interquartile range) for continuous measures and N (percentage) for categorical measures. Differences between baseline and postKTx were assessed using paired-sample *t*-test or Wilcoxon matched-pairs signed rank test for continuous variables, while McNemar’s test was used for categorical variables. Significant *p*-values (*p* < 0.05) are depicted in **bold**.

**Table 2 ijms-24-03640-t002:** Univariate correlations between the baseline serum concentrations of uremic toxins indoxyl sulphate and p-cresyl sulphate and the functional and biochemical markers of endothelial function or vessel stiffness.

		Indoxyl Sulphate	p-Cresyl Sulphate
RHI (Baseline)	*ρ*	−0.329	−0.214
	*p-*value	0.101	0.293
RHI (postKTx)	*ρ*	−0.461	0.053
	*p-*value	**0.018**	0.798
AIx (Baseline)	*ρ*	−0.208	0.036
	*p-*value	0.308	0.862
AIx (postKTx)	*ρ*	−0.176	0.075
	*p-*value	0.389	0.717
P-selectin (Baseline)	*ρ*	0.118	−0.116
	*p-*value	0.573	0.580
P-selectin (postKTx)	*ρ*	0.510	0.122
	*p-*value	**0.009**	0.560
MMP-9 (Baseline)	*ρ*	0.017	−0.054
	*p-*value	0.933	0.792
MMP-9 (postKTx)	*ρ*	0.201	0.236
	*p-*value	0.326	0.245

postKTx, post-transplantation; AIx, augmentation index; MMP-9, matrix metalloproteinase 9; RHI, reactive hyperemia index; SBP, systolic blood pressure. Univariate correlations were assessed using Spearman’s rank correlation. Significant *p*-values (*p* < 0.05) are depicted in **bold**.

**Table 3 ijms-24-03640-t003:** Multivariate linear regression analyses between indoxyl sulphate and reactive hyperemia index and P-selectin (dependent variables), both at two years post-transplantation, adjusted for age, sex, presence of diabetes, dialysis treatment and statin usage.

Independent Variables		RHI postKTx	P-Selectin postKTx (ng/mL)
Indoxyl sulphate (µmol/L)	**ꞵ**	−0.506	0.518
	***p*-value**	**0.028**	**0.036**
Age (years)	**ꞵ**	0.271	−0.181
	***p*-value**	0.195	0.417
Sex (male)	**ꞵ**	−0.109	−0.025
	***p*-value**	0.586	0.908
Presence of diabetes (yes/no)	**ꞵ**	−0.032	−0.135
	***p*-value**	0.870	0.532
Dialysis treatment (yes/no)	**ꞵ**	0.319	0.073
	***p*-value**	0.139	0.748
Statin usage (yes/no)	**ꞵ**	0.056	0.243
	***p*-value**	0.804	0.324

## Data Availability

The data underlying this article cannot be shared publicly due to ethical reasons, e.g., for the privacy of individuals that participated in the study. The data will be shared on reasonable request to the corresponding author.

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
