# Peer review of "Indoxyl Sulphate Retention Is Associated with Microvascular Endothelial Dysfunction after Kidney Transplantation"

_ijms, 2023, doi:10.3390/ijms24043640_

Round 1

Reviewer 1 Report

This is a very inetersting paper, with novel results and very well presented 

Reviewer 2 Report

1. The study is of interest; however, the length of this manuscript is short.

2. It is highly recommended to publish this manuscript in the form of "communication".

Reviewer 3 Report

In this study, the authors aimed to investigate if functional properties of the vasculature differed two-years post-KTx (postKTx) when compared to baseline (time 19 of KTx). The concluded that IS promotes worsened endothelial dysfunction post-KTx, which may contribute to the sustained CVD risk post-KTx.

The overall level of the manuscript is very good which is investigated an interesting topic, apart from lack of clarity in some places, and some specifics. However, ther is a list of comments that will help the audience to appreciate the advancement of the presented work:

1. The manuscript requires proofreading and revision to improve the quality of English. 

2. The authors should follow the instructions of the Journal while they preparing the figures and tables and thier Captions, likewise the refernces. 

3. Statistical analysis needs to be revised by an expert in the field.

4. Discussion needs to be strengthened.

5. Figure S1 could be added in the same MS.

Reviewer 4 Report

The manuscript “Indoxyl Sulphate Retention is Associated with Microvascular Endothelial Dysfunction After Kidney Transplantation”, by Hobson and collaborators deals with clinically important issue of mechanisms underlying the cardiovascular disease post kidney transplantation. Although the paper is well organized and clearly written and is easy to read it, there are some issues that need to be corrected in order for the work to be suitable for publication. A manuscript fits into the scope of the journal. Hence, the manuscript could be interesting for readers and I recommend to reconsider it for publication after major revision.

Issues for improvement:

- The title should be corrected, since the experimental design did not provide direct evidence that indoxyl sulphate is responsible for endothelial disfunction after kidney transplantation, nor direct evidence that it is responsible for microvascular disfunction (only correlation in study in vivo was found; ex vivo, arteries were studied, not a microcirculation).

Hence, in general authors should be more cautious in conclusions about direct relation.

- Table 1 revealed substantial increase in percentage of statin and ACEi/ARB users, and decrease in beta blocker users.  It says that these differences were not significant, but it does not seem reasonable concerning numbers and I do not see which statistical method was used to prove that?  Only chi-squared test would be appropriate to make a comparison of two proportions – please provide results of the chi-square test and detail explanation how it might influence the result in discussion section. Also a comparison of the results according to the use of particular medications is needed.

- Figure 1. needs some corrections:

 – “wire myography experiments to test effects on”- it is missing on what!?

 - quantification of serum Indoxyl Sulphate and p-Cresyl Sulphate was not made post transplantation

-  discussion: “higher IS levels at baseline were predictive of worsened 251endothelial function postKTx.” Regression analysis was not done, hence prediction was not proved, only correlation.

- Material and methods: please explain the meaning of “by inflating the blood pressure cuff to a supra-systolic pressure of 60 mmHg or 200 mmHg (whichever was higher).” Was it systolic, diastolic – what was 60mmHg? What does it mean “whichever was higher”?

- Material and methods: blood specimens – please provide details about blood specimens collection and preservation prior the analysis and some details on analysis (methods, analyzers…)

- Material and methods: Please provide detail explanation about blood vessel sampling – which vessels, how they are isolated, under which conditions, what were the diameters of vessels used. Also data about analyses are missing – number of experiment repetitions, how many preparations were obtained from one vessel/tissue sample…

Reviewer 5 Report

The introduction/background section and the discussion section needs to be improved to stress the significance of the paper. It is also important to highlight the exact background about what IS is and lay out the hypothesis clearly. A more extensive literature analysis in the background section or maybe a reflection during discussion, which describes the signalling pathways that have been speculated to be a link between CKD and CVD would be useful. A model should be included to explain the authors findings cause the discussion isn't easy to read how it is currently.

Round 2

Reviewer 3 Report

 The paper now can be accepted without any further changes.

Author Response

We thank the reviewer for taking their time to read our manuscript and give constructive feedback.

Reviewer 4 Report

The manuscript “Indoxyl Sulphate Retention is Associated with Microvascular Endothelial Dysfunction After Kidney Transplantation”, was partially improved according to my remarks.

Issues for further improvement:

Original remark: “- Table 1 revealed substantial increase in percentage of statin and ACEi/ARB users, and decrease in beta blocker users.  It says that these differences were not significant, but it does not seem reasonable concerning numbers and I do not see which statistical method was used to prove that?  Only chi-squared test would be appropriate to make a comparison of two proportions – please provide results of the chi-square test and detail explanation how it might influence the result in discussion section. Also a comparison of the results according to the use of particular medications is needed.

Authors response:

“ Indeed, this is a mistake on our behalf. In fact, when consulting our biostatistics expert, we were informed McNemar’s test should be implemented to assess differences between categorical variables with repeated measures. After repeating the statistical analysis, a significant difference was observed for ‘Statin users’ and ‘Beta blocker users’ between baseline and post-transplantation.

We also thank the reviewer for their second suggestion to conduct subgroup analysis. However, after careful consideration, we conclude that subgroup analysis is not possible in our cohort due to the following reasons:

           While we agree that investigation of potential differences according to use of medication would be of interest, disentangling the impact of taking or not taking a particular medication is not possible with a small sample size (n=27) and information collected for three medications. After stratifying patients by statin usage, significant differences would only suggest statins may impact reactive hyperaemia index or augmentation index if all patients were treated with ACEi/ARBs, or no patients were treated with ACEi/ARBs. After careful review of our data, conducting such analyses was not possible.

           In the clinical context, the potential effects of medication in our study are somewhat more complex than simply drawing conclusions from subgroup analyses. We cannot rule out the possibility of ACEi/ARBs influencing functional properties before or after kidney transplantation. This again makes subgroup analysis difficult to implement, since the timepoint at which to stratify is unclear.

           We assessed patients in a repeated cross-sectional manner, i.e., at baseline and two years post transplantation. As a result, we cannot detail how long a patient was treated with a particular medication. For example, 23/27 (85%) of patients were treated with statins at follow-up but for how long is unknown.

           Lastly, stratifying patients by use of medication at follow-up, which seems most intuitive, would not be suitable for use of statins due to the large % of patients on this medication. Subgroup analysis for 23 vs 4 patients to investigate differences would then lack statistical power.

Differences between clinical parameters at baseline and follow up may explain observed changes to vessel stiffness and endothelial function. We feel this statement is suitable without subsequent subgroup analysis. Therefore, we included statin and beta blocker usage in the following statement, (refer to line 290):

“We observed significantly reduced calcium, phosphate, lipoprotein(a), systolic/diastolic blood pressure and beta blocker usage, as well as increased HDL cholesterol and statin usage after postKTx – all of which may partially explain, or explain in composite, observed changes to endothelial function and/or vessel stiffness, however further and larger studies are required for clarification.” “

Thank you very much for your effort. However, I do not agree that at least regression analyses could not be done (instead of stratification) for medication usage. In addition to this analysis all of the explanations from the above should be copied as a main limitation of the study.

Author Response

We thank the reviewer for their response. While we agree that stratification is not possible for the reasons given, further adjustment for medication in our multivariate linear regression is necessary. In light of this, we need to acknowledge that lipids (and lipid-lowering treatment) are related to uremic toxins and could further introduce some statistical bias. This was recently reported by our group (doi: 10.3390/toxins14060412) and should be considered when interpreting the results.

Following your comments, we decided that baseline data should be used to model the independent effects of statins/blood pressure-lowering medication since indoxyl sulphate (IS) was also measured at baseline. Instead of adjusting for medications within the same class of drug, a blood pressure-lowering medication variable was made by combining beta blockers, RAAS blockers (ACEi and ARBs), calcium channel blockers and diuretics. All 27 (100%) patients at baseline were using a form of blood pressure-lowering medication, therefore this variable was not included in the multivariate linear regression model. 10 (27%) of patients were taking statins (Lipid-lowering medication), hence this variable was included in the models. After adjustment for age, sex, dialysis, presence of diabetes and statin usage, both previously significant associations remained significant:

  • IS was independently, negatively associated with RHI postKTx (standardized beta = -0.506; P < 0.05).
  • IS was independently, positively associated with P-selectin postKTx (standardized beta = 0.518; P < 05).

Therefore, based on the new analyses, we made the following adjustments to the manuscript:

  • Abstract modified to: “Furthermore, baseline serum indoxyl sulphate (IS), but not p-cresyl sulphate, was independently, negatively associated with reactive hyperaemia index, a marker of endothelial function, and independently, positively associated with P-selectin postKTx.”
  • ACEi/ARBs and beta blocker rows were replaced with blood pressure-lowering medication in Table 1.
  • The follow sentence was added to the Table 1 legend: “Blood-pressure-lowering medication includes use of beta-blockers, diuretics, angiotensin-converting enzyme inhibitors, angiotensin II receptor blockers and/or calcium channel blockers.”
  • To describe changes to multivariate linear regression models, the following text was added to the Results section: “To identify independent associations between uremic toxins and markers of endothelial function or vessel stiffness, we performed multivariate linear regression analyses with adjustment for age, sex, presence of diabetes, dialysis treatment and statin usage, only for those parameters that were significantly correlated to IS or pCS in univariate analyses. Our analyses revealed that IS was independently negatively associated with RHI postKTx (standardized beta = -0.506; P < 0.05). IS was also independently positively associated with P-selectin postKTx (standardized beta = 0.518; P < 0.05).”
  • The following sentence was added to Table 3’s figure legend: “Blood pressure-lowering medication was not included since all patients were on this treatment at baseline.”
  • The following sentence was added to Methods section 4.3 legend: “Blood-pressure-lowering medication includes use of beta-blockers, diuretics, angiotensin-converting enzyme inhibitors, angiotensin II receptor blockers and/or calcium channel blockers.”

Lastly, we addressed the problem surrounding our inability to stratify groups by use of medication in the Discussion. We listed this as a an important limitation of our study. The following text was added: “Lastly, we were unable to stratify patients by use of medication due to the small sample size and only collecting data at two time points. We were, however, able to partly resolve this issue by adjusting for statin use at baseline in our multivariate linear regression models.”

Round 3

Reviewer 4 Report

Authors managed to respond appropriately to my remarks and the manuscript is substantially improved.